# Synergistic biodegradation of aromatic-aliphatic copolyester plastic by a marine microbial consortium

Ingrid E. Meyer-Cifuentes [1], Johannes Werner [2], Nico Jehmlich [3], Sabine E. Will[4], Meina Neumann-Schaal [4] & Başak Öztürk [1✉]

The degradation of synthetic polymers by marine microorganisms is not as well understood as the degradation of plastics in soil and compost. Here, we use metagenomics, metatranscriptomics and metaproteomics to study the biodegradation of an aromatic-aliphatic copolyester blend by a marine microbial enrichment culture. The culture can use the plastic film as the sole carbon source, reaching maximum conversion to $CO_2$ and biomass in around 15 days. The consortium degrades the polymer synergistically, with different degradation steps being performed by different community members. We identify six putative PETase-like enzymes and four putative MHETase-like enzymes, with the potential to degrade aliphatic-aromatic polymers and their degradation products, respectively. Our results show that, although there are multiple genes and organisms with the potential to perform each degradation step, only a few are active during biodegradation.

[1] Junior Research Group Microbial Biotechnology, Leibniz Institute DSMZ-German Collection of Microorganisms and Cell Cultures, Inhoffenstraße 7B, 38124 Braunschweig, Germany. [2] Department of Biological Oceanography, Leibniz Institute of Baltic Sea Research, Seestraße 15, D-18119 Rostock, Germany. [3] Department of Molecular Systems Biology, Helmholtz-Centre for Environmental Research-UFZ, Permoserstraße 15, 04318 Leipzig, Germany. [4] Junior Research Group Bacterial Metabolomics, Leibniz Institute DSMZ-German Collection of Microorganisms and Cell Cultures, Inhoffenstraße 7B, 38124 Braunschweig, Germany. ✉email: basak.oeztuerk@dsmz.de

Three fifty million tons of plastics were produced worldwide exclusively in 2017[1]. Over 250,000 tons of them were released to and found floating in the oceans[2,3]. In 2015, it was estimated that by 2025 the plastic waste in the ocean could increase by an order of magnitude[4]. Commercial production of biodegradable plastics started in the early 2000s[5] as an ecologically friendlier alternative to conventional plastics[6–8]. The manufacturing of these plastics are intended for specific applications, such as items with a shorter life time and where recycling is not feasible[9,10]. Most biodegradable polymers contain ester bonds that are susceptible to enzymatic hydrolysis[9,11,12]. Poly(butylene adipate-co-terephthalate) (PBAT) is a commercial aromatic-aliphatic copolyester composed of adipic acid (Ad), terephthalic acid (Te), and 1, 4-butanediol (B). It has favorable mechanical properties due to its aromatic components, in combination with biodegradability granted by the more flexible aliphatic components[13,14].

The depolymerization of PBAT by soil and compost microorganisms is well-studied: known enzymes with PBAT hydrolytic activity include cutinase-like serine hydrolases[15–17], mostly originating from terrestrial *Actinomycetes* and fungi. So far, only one PBAT-hydrolyzing enzyme has been characterized from the aquatic environment[14]. The enzymatic hydrolysis of PBAT yields a mixture of the terephthalate-butanediol monoester (BTe), and the monomers[16]. PBAT-degrading enzymes have different degrees of activity toward the BTe intermediate. The PBAT-degrading enzyme Ppest from *Pseudomonas alcaligenes* cannot degrade BTe, which almost completely inhibits the enzyme's activity[14]. Similarly, the cutinase from *Humicula insolens*, HiC, has a much lower activity on BTe than the Thc_Cut1 from *Thermobifida cellulosilytica*, which can degrade it efficiently to monomers[16]. Therefore, it was proposed that an esterase with a high specificity to the BTe could aid enzymatic PBAT hydrolysis by removing this inhibitory degradation product[14]. This proposed degradation pathway is similar to the PET-degradation pathway by *Ideonella sakaiensis*[18]. The degradation is initiated by PETase, an $\alpha/\beta$ hydrolase that degrades the polymer into the monoester mono-2-hydroxyethyl terephthalate (MHET). This monoester is hydrolyzed by another $\alpha/\beta$ hydrolase, MHETase, to form Te and ethylene glycol[18]. *I. sakaiensis* also carries a Te degradation (TPD) cluster, which converts Te into protocatechuate (PCA) in a two-step reaction[18,19]. After conversion into pyruvate and oxaloacetate by the PCA degradation (PCD) cluster, the aromatic monomer is completely metabolized[19]. Due to the structural similarity of PBAT and PET, some PBAT-degrading enzymes can also degrade PET, albeit more slowly[17,20]. It is so far unknown whether a two-step degradation pathway for PBAT with an intermediary monoester-cleaving enzyme exists.

Regardless of whether their enzymes can hydrolyze the polymer to its monomers or not, most PBAT-degrading microorganisms discovered to date cannot use the monomers as C source, and therefore are not able to degrade the polymer into biomass and $CO_2$. It has been stated that in case of a mature compost environment, other members of the microbial community can metabolize the released monomers[21]. It is possible that such complex synthetic polymer blends would be mineralized by consortia rather than single microorganisms in the environment. Mechanisms of cooperative degradation however have not yet been investigated. Due to the structural similarity of PBAT degradation intermediates to those of PET, we hypothesized that the complete PBAT degradation by a microbial consortium may follow a similar pathway to PET degradation by *I. sakaiensis*.

In this study, we investigate the mineralization of a commercial PBAT-based blend film (PF) by a marine enrichment culture. PF is composed of PBAT blended with a copolyester of Te, B, and sebacic acid (Se) (instead of Ad), and also contains a small amount of polylactic acid. In this framework, we elucidate the degradation pathway of this biodegradable plastic with an integrated metaomic approach. The abundance and distribution of individual microorganisms in the consortium, their potential roles in the PF degradation process, and the relation to catabolic genes from other environments are investigated. Our study describes a marine microbial consortium that synergistically degrades a complex biodegradable copolymer and proposes a mechanism for the biodegradation.

## Results

We enriched a marine microbial community in an artificial marine medium supplemented with PF as sole carbon source (named I1 culture). To elucidate which microorganisms and genes play a role in PF biodegradation, three independent experiments were carried out (Fig. 1): the first experimental setup

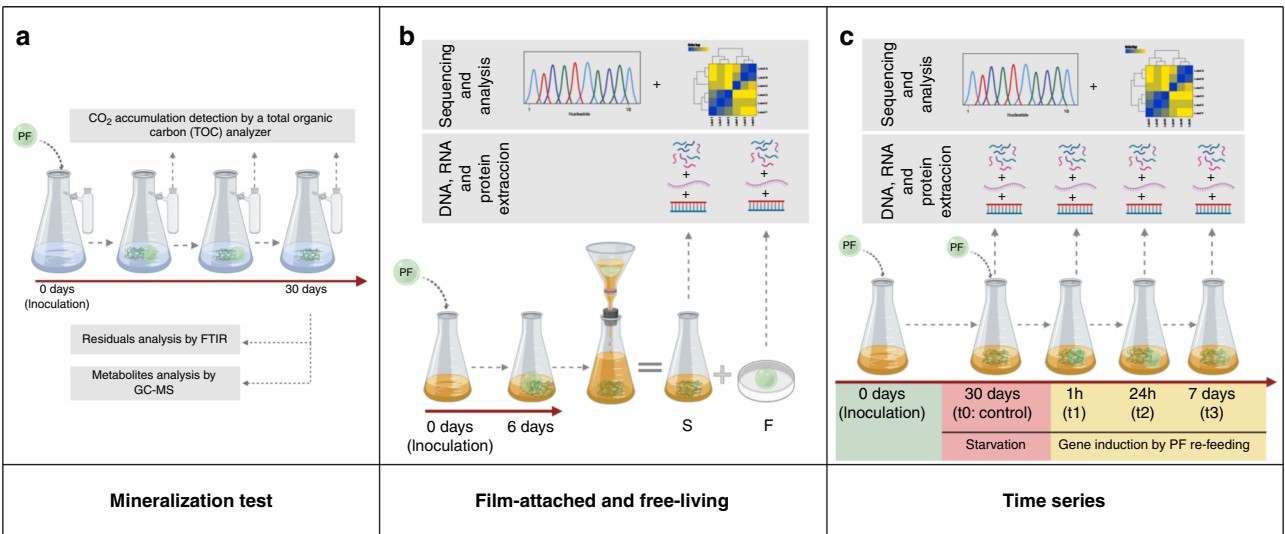

**Fig. 1 Experimental setup. a** Analysis of plastic-carbon use by measuring $CO_2$ production in the presence of PF, (**b**) diversity of biofilms tested on PF-attached (F) against free-living bacteria (S), (**c**) gene expression and protein biosynthesis of catabolic genes during different time points (t1, t2, t3) relative to a control (t0) in the presence of PF. **a**, **b**, **c** experiments were performed independently from each other and each experiment consisted of three biological replicates (created with Biorender.com).

(a) aimed to detect biodegradation products and $CO_2$ production due to microbial activity. In the second experiment (b) we aimed to analyze differences between the film-attached (F) and free-living bacteria (S) through metaomics. Finally a time series experiment (c) was performed to identify putative genes and proteins needed for the biodegradation of PF through metaomics.

**Mineralization of the PF and monomers**. The mineralization tests were carried out for 30 days to ensure complete biodegradation (Fig. 1a). However, disintegration of PF was already visually observed after six days. Within a month, three independent experiments showed that the communities converted around 60% (Fig. 2a (a1, a2, a3)) of the initial carbon content supplied in the films into $CO_2$. The highest degradation rate was registered between days 6 and 10 (11–16% mineralized/day), reaching a plateau after 15 days. Although most of the film seemed to be disintegrated, some solid residues remained in these cultures.

Release of PF oligomers and monomers, and the composition of solid residues remaining after complete mineralization were further analyzed. Metabolites were extracted from the cells and cell-free culture supernatants after eight and twenty days of incubation with PF. After eight days of incubation, the cell biomass contained low amounts of the monomers Ad, Se, and Te, as well as the monoester BTe. After twenty days, the intracellular amounts of Ad and Te were 95% reduced and Se and BTe were depleted below detection limit (Table S1). Monomers and oligomers were undetectable in the extracellular environment (data not shown). In parallel, the solid residues remaining after PF degradation were collected, analyzed by FTIR and contrasted to plastic samples incubated in the absence of the I1 culture (Fig. S1). It was observed that the main ester peak of PF at 1710–1750 $cm^{-1}$ disappeared completely in most of the biological replicates. Peaks corresponding to the solid $CaCO_3$ mineral filler were detected in all samples, as well as those that potentially belong to biofilm components (detailed peak analysis is given in Fig. S1). The plastic samples that were incubated in the culture medium for 30 days without the addition of the I1 culture did not show any change in the FTIR spectra (Fig. S1a). Based on the FTIR spectra, we conclude that the remaining solid residues are $CaCO_3$ mineral filler with attached biofilm, and almost all of the polyester was degraded.

Before PF disintegration, the biofilm-forming ability of the community was analyzed. Scanning electron microscopy (SEM) imaging revealed that the PF surface was colonized by the I1 community already after three days (Fig. 2b). At this time, we observed pits distributed uniformly across the PF surface resulting from microbial activity. After six days, larger holes and craters were formed. Microbial biofilm surrounded by exopolysaccharides resided in these holes. The main microbial morphotype were rod shaped cells of about 2 μm. At this time, the film was very fragile and began to disintegrate.

In addition to the mineralization of the film itself, the mineralization of each monomer was measured separately. The maximum conversion to $CO_2$ was 50%, 59%, 29%, and 40% for B, Te, Se, and Ad acid, respectively (Fig. S2). The $CO_2$ production reached a plateau after 10 days.

**Taxonomic and functional profiling of the I1 community**. In the following section, the results of the bioinformatic analyses of the experiments as illustrated in Fig. 1b and c are described. For each experiment, DNA, RNA, and proteins were extracted from the same sample. Metagenomes were assembled, binned and taxonomically classified. The reads were mapped to the bins for abundance estimation. Metagenome binning yielded 32 bins of

>95% completeness and <10% contamination (Table S2). Taxonomic profiling revealed that the I1 culture is a diverse community composed mainly of Alphaproteobacteria, Gammaproteobacteria and Flavobacteria, as well as Actinobacteria albeit in lower numbers. The abundance profiles remained stable throughout the time series experiment (Fig. 3a). The six most abundant bins in this experiment were three of the family Rhodobacteraceae, i.e., bin 17 and 20 (*Pseudooceanicola* spp.), bin 10 (unclassified Rhodobacteraceae bacterium), one *Marinobacter* (bin 32), one *Aequorivita* (bin 1) and one of the family Micavibrionaceae (bin 3). These bins comprised around 80% of the binned population (Fig. S3).

The microbial community composition of the biofilm on the plastic surface was contrasting to the free-living community composition (experimental setup illustrated in Fig. 1b). Relative abundances of Alphaproteobacteria (bins 2, 13, 14, 17, 18, 23, 26, 27) strains and a few Bacteroidia (bin 1, 30), Actinobacteria (bin 6) and Gammaproteobacteria (bin 12) strains increased in the free-living fraction compared to film-attached communities (Fig. 3a, Fig. S4). In contrast, the Gammaproteobacteria *Marinobacter* sp. (bin 32) and Marinicellaceae (bin 28), the Bacteroidium *Geldibacter* sp. (bin 16), the Acidomicrobium *Ilumatobacter* (bin 4), Phycisphaerae (bin 5), and an Alphaproteobacterium belonging to Micavibrionaceae (bin 3) were enriched in the film-attached fraction. These observations were supported by correlating their abundance distribution. The abundances of the film-associated taxa (bins 3, 5, 16, 28, 32) are significantly positively-correlated to each other. Similarly, the abundances of free-living taxa (1, 2, 9, 10, 12, 13, 14, 15, 17, 21, 23, 26, 30, 33) are positively correlated with each other. Abundances of film-associated bins are negatively-correlated to the abundances of most free-living taxa (Fig. 3b).

Overall, the four most abundant bins made up 75–80% of the binned population in both film-attached and free-living fractions (Fig. S3). This scenario changed when the monomers were used as sole carbon sources instead of PF. When B was given as sole carbon source, *Marinobacter* sp. (bin 7) was the most abundant bacterium, comprising 80% of the binned community. Similarly, *Saccharospirillum* sp. (bin 12) dominated the cultures with Te, comprising 80% of the binned community (Fig. S5). Both bacteria were present in low amounts (≤2%) when the culture was growing with PF as carbon source. The addition of Ad and Se triggered the growth of multiple bacteria, where bins 1, 13 and 32 were the most abundant, comprising 65% of the community in both cases. Interestingly, bins 10 and 17, which were among the most abundant bacteria throughout the time series and the free-living community, were barely traceable when single monomers were given as carbon source (Fig. S3).

**Genomic context of the putative PF depolymerases**. The functional analysis of the metagenomes revealed a diverse gene repertoire which is potentially related to polymer degradation. Six orthologues of the three known PETases (named as PETase-like enzymes (Ples)); *I. sakaiensis* PETase (A0A0K8P6T7, GAP38373)[18] (IsPETase), leaf compost cutinase (AEV21261.1)[22] and *Thermobifida fusca* cutinase (ADV92528.1)[23] were recovered from the two *Marinobacter* bins 21 (Ple200 and Ple201) and 32 (Ple611, Ple453, Ple628 and Ple629) (Table 1). The Ples show a 45–50% amino acid (aa) identity to the known PETases except Ple453 (29–32%) (Table 1). All Ples carry the common lipase/esterase consensus motif G-X1-S-X2-G, the IsPETase catalytic triad His237, Ser160, and Asp206[24] (Fig. S6a) and were classified as α/β hydrolases[25]. All proteins carry signal peptides for periplasmic localization via the Sec pathway. Similar to the tandem cutinases *est1* and *est119* of *Thermobifida alba* AHK119[26], the genes

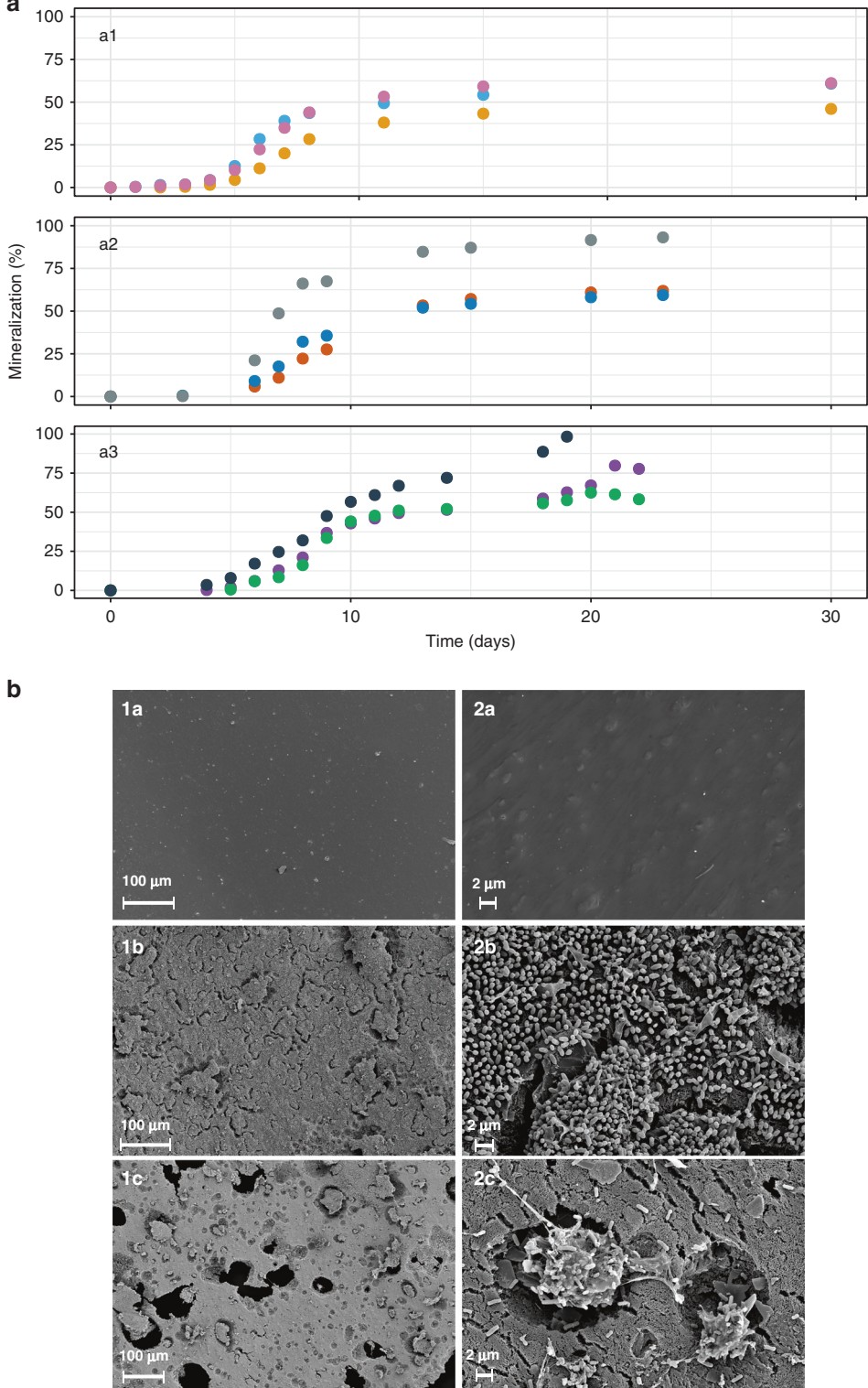

**Fig. 2 Microbial growth on PF. a** Mineralization and $CO_2$ production of PBAT-based blend film (PF) by the consortia I1. Three biological replicates of three independent tests (a1, a2, a3) are shown. Each color depicts a biological replicate. **b** SEM image of control (1a, 2a), biofilm formation and plastic degradation after three days (1b, 2b) and six days (1c, 2c). The experiment was performed in two repetitions with a similar outcome. Source data are provided as a Source Data file.

encoding for Ple200 and Ple201, as well as Ple628 and Ple629 have a tandem structure. In both cases, the genes were separated by 359 nucleotides (nt) without interjacent ORFs. The intergenic spaces of each tandem pair had different sequences. These both contained putative transcription termination sites, indicating that

the genes may be independently transcribed. Ple200 and Ple201 from bin 21 are very closely related to Ple628 and Ple629 from bin 32, with an aa identity of 90% between Ple200 and Ple628 or Ple201 and Ple629, respectively. The aa identity between each tandem pair is 74–75%. The 71 kbp contig with the *ple* genes in

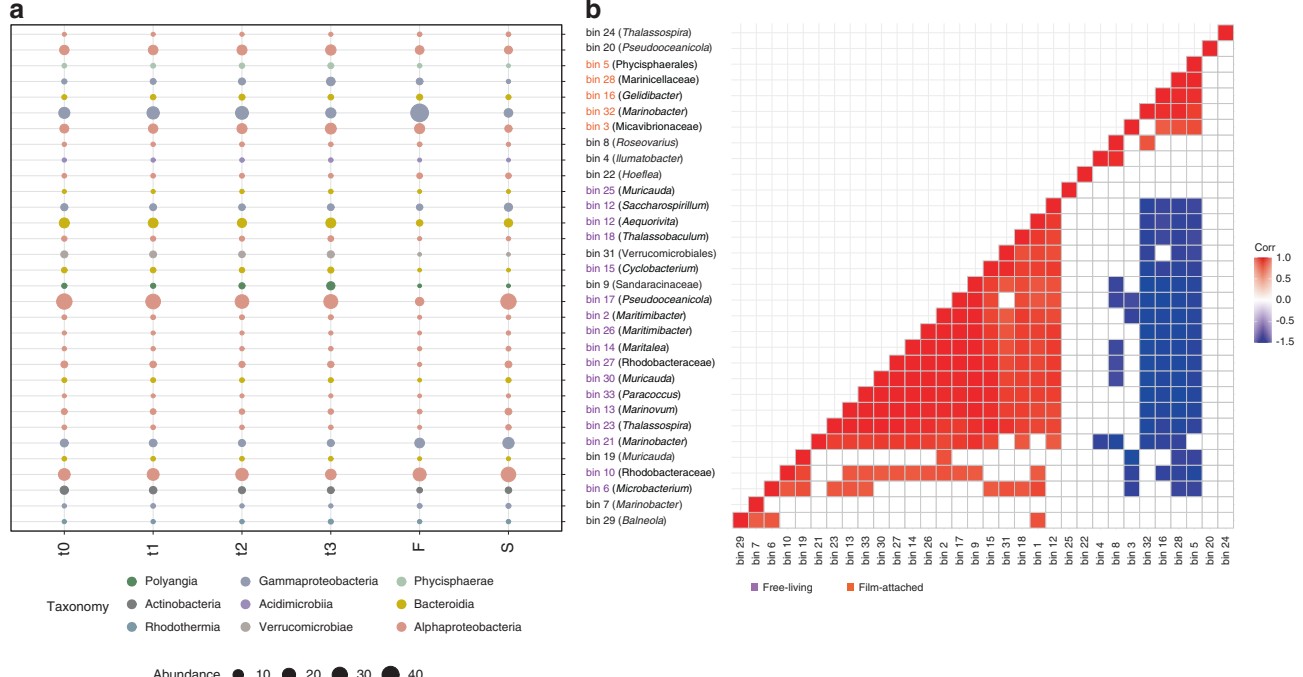

**Fig. 3 Bins profiling.** Abundances were calculated by counting the mapped reads per bin. Percent abundances were calculated based on the number of mapped reads per metagenome. The percentage of unbinned reads per replicate metagenome can be found in Table S3. **a** Taxonomic abundance profile during time series (t0, t1, t2, t3) in the presence of PF, film-attached (F) and free-living cells (S) is shown. The size of the bubbles represent the average relative abundance of each bin in the I1 population. The color of each bubble represents a taxa. The bins belonging to the free-living and film-attached bacteria fractions are highlighted in purple and red, respectively. The percentages for monomer cultures are averages of biological duplicates and the rest are the average of biological triplicates. **b** Correlation between the percent abundances of film-attached and free-living bacteria. Negative correlations are shown in shades of blue, positive correlations are shown in shades of red. Percent abundance data for each biological replicate was used as input. Source data are provided as a Source Data file.

bin 21 has a similar sequence (85% nt identity) and high synteny to the corresponding region in bin 32. Most Ples clustered phylogenetically together with related genes from two other *Marinobacter* species, Ple453 being more distantly related to these, as well as to the known PETases (Fig. S7a).

**Genomic context of the putative downstream degradation genes.** In addition to the Ples, four genes encoding MHETase-like enzymes (Mles) were recovered from the metagenome (Table 1). These have a 29–47% aa identity to the *I. sakaiensis* MHETase (A0A0K8P8E7) and carry the MHETase catalytic triad[27] (Fig. S6b). Like Ples, Mles carry a signal Sec peptide. Three of them could be allocated to a certain host: Mle800, Mle267, and Mle288 (Table 1). Mle267 and Mle800 have a 60% aa identity to each other. They are however distantly related to both Mle046 and Mle288 (Fig. S7b).

The fourth Mle, Mle046, has the highest identity to the IsMHETase (Table 1). The *mle046* gene is 100% identical to the homolog present on the *Celeribacter manganoxidans* plasmid pDY25-B (CP021406.1). In the I1 metagenome, *mle046* is located on an unbinned 107 kbp contig (contig 279). This contig also carries a conjugation-related operon, and thus possibly belongs to a conjugative plasmid. This operon is closely related (>90%) to those from several conjugative plasmids from marine Alphaproteobacteria, including pDY25-B (92% nt identity). On pDY25-B, the *mle046* homolog and eight other ORFs are flanked on either side by a Tn7 transposase, forming a putative composite transposon. The *mle046* is located at the end of contig 279, and is also flanked by a Tn7-type transposase on the right side. The two left-flanking ORFs, as well as the left-flanking transposase which are present on the pDY25-B transposon are

missing from this contig. The rest or this transposon is almost identical (99% nt identity) to the pDY25-B transposon (Fig. S8). The genomic context of this unbinned Mle therefore suggests that it is located on an Alphaproteobacterial conjugative plasmid, and has been acquired via a transposable element. There was no evidence of acquisition via horizontal gene transfer for the other Mles.

Two different TPD clusters were found in the I1 metagenome (Table 1). These were located on bin 12 (named sTPD), as well as the unbinned 21 kbp-contig 1092 (named pTPD). Both clusters are orthologous to those from *Comamonas* sp. E6[19] and *I. sakaiensis*[18] (Table 1). The pTPD cluster lacks the terephthalate permease gene *tphC*[28] which is present in *Comamonas* and *Ideonella* (Fig. S9). No homologs of the *tphC* were present in the I1 metagenome. The pTPD cluster has a 100% nt identity to the TPD cluster on pDY25-B. The contig 1092 carries a gene encoding for the plasmid partitioning protein ParA with a 85% nt identity to the *parA* on the *Phaeobacter piscinae* plasmid pP18i (CP010724.1). The genomic context of the unbinned pTPD cluster indicates that, like *mle046*, the pTPD cluster may be part of an Alphaproteobacterial plasmid. The sTPD cluster has a different structure to those of *Comamonas* and *Ideonella*, as it only contained the *tphA2A3B* genes, missing the reductase component *tphA1* (Table 1). Adjacent to the sTPD genes, homologs of *pht2345* genes for phthalate degradation[29] are present (Fig. S9). The abundance of bin 12 when Te is added as sole C source indicates that this organism can use Te as C source, despite the missing *tphA1*. Phylogenetically, TphA2 subunits from I1 form three distinct clades: pTphA2 (Tpad1092), sTphA2 (Tpad12), and *Ideonella/Comamonas* TphA2 and relatives clustering separately from each other (Fig. S7c).

**Table 1 Identification of PF degradation genes.**

| Protein ID | Function | Bin | Genus | Nearest relative | % aa identity |
|---|---|---|---|---|---|
| Ple611 | Hydrolase (PETase-like) | 32 | *Marinobacter* | *Thermobifida cellulosilytica* cutinase (E9LHV9) | 50.0 |
| Ple628 | Hydrolase (PETase-like) | 32 | *Marinobacter* | *Ideonella sakaiensis* PETase (A0A0K8P6T7) | 50.5 |
| Ple629 | Hydrolase (PETase-like) | 32 | *Marinobacter* | *I. sakaiensis* PETase (A0A0K8P6T7) | 53.2 |
| Ple200 | Hydrolase (PETase-like) | 21 | *Marinobacter* | *I. sakaiensis* PETase (A0A0K8P6T7) | 49.2 |
| Ple201 | Hydrolase (PETase-like) | 21 | *Marinobacter* | *I. sakaiensis* PETase (A0A0K8P6T7) | 49.1 |
| Ple453 | Hydrolase (PETase-like) | 32 | *Marinobacter* | *Thermobifida fusca* cutinase (E9LHV10) | 28.3 |
| Mle800 | Hydrolase (MHETase-like) | 32 | *Marinobacter* | *I. sakaiensis* MHETase (A0A0K8P8E7) | 31.0 |
| Mle046 | Hydrolase (MHETase-like) | unknown | unknown | *I. sakaiensis* MHETase (A0A0K8P8E7) | 46.9 |
| Mle288 | Hydrolase (MHETase-like) | 12 | *Saccharospirillum* | *I. sakaiensis* MHETase (A0A0K8P8E7) | 38.6 |
| Mle267 | Hydrolase (MHETase-like) | 17 | *Pseudooceanicola* | *I. sakaiensis* MHETase (A0A0K8P8E7) | 29.3 |
| pTphA1 | Terephthalate 1,2-dioxygenase reductase | unknown | unknown | *Comamonas* sp. TphA1 (Q3C1E0) | 37.1 |
| pTphA2 | Terephthalate 1,2-dioxygenase oxygenase large subunit | unknown | unknown | *Comamonas* sp. TphA2 (Q3C1E3) | 60.2 |
| pTphA3 | Terephthalate 1,2-dioxygenase oxygenase small subunit | unknown | unknown | *Comamonas* sp. TphA3 (Q3C1E2) | 43.2 |
| pTphB | Terephthalate dihydrodiol dehydrogenase | unknown | unknown | *Comamonas* sp. TphA1 (Q3C1E1) | 40.5 |
| pTphR | Terephthalate 1,2-dioxygenase transcriptional regulator | unknown | unknown | *Comamonas* sp. TphR (Q5D0X8) | 38.1 |
| sTphA2 | Terephthalate 1,2-dioxygenase oxygenase large subunit | 12 | *Saccharospirillum* | *Comamonas* sp. TphA2 (Q3C1E3) | 60.2 |
| sTphA3 | Terephthalate 1,2-dioxygenase oxygenase small subunit | 12 | *Saccharospirillum* | *Comamonas* sp. TphA3 (Q3C1E2) | 48.3 |
| sTphB | Terephthalate dihydrodiol dehydrogenase | 12 | *Saccharospirillum* | *Comamonas* sp. TphA1 (Q3C1E1) | 44.7 |
| pht2 | Phthalate 4,5-dioxygenase oxygenase reductase | 12 | *Saccharospirillum* | *Pseudomonas putida* Pht2 (Q05182) | 46.4 |
| pht4 | 4,5-dihydroxyphthalate dehydrogenase | 12 | *Saccharospirillum* | *P. putida* Pht4 (Q05184) | 50.5 |
| pht5 | 4,5-dihydroxyphthalate decarboxylase | 12 | *Saccharospirillum* | *P. putida* Pht5 (Q59727) | 73.9 |
| pht3 | Phthalate 4,5-dioxygenase oxygenase | 12 | *Saccharospirillum* | *P. putida* Pht3 (Q05183) | 62.0 |

The Uniprot ID of each nearest relative is indicated in the parenthesis.

The conversion of PCA to pyruvate and oxaloacetate for complete terephthalate degradation can be accomplished via PCA 4,5-[19,30,31], PCA 2,3-[32] or PCA 3,4-[33] cleavage. In the I1 metagenome, diverse protocatechuate 4,5- and 2,3-dioxygenase subunits were found. These, with the exception of those from *Saccharospirillum* all belonged to Alphaproteobacteria (Table S4).

**Gene expression and protein biosynthesis during the time series**. The metatranscriptome and -proteome of the I1 community was characterized following the addition of fresh PF substrate after a starvation period (Fig. 1c). At each sampling point, about 70% of all genes were detected (cutoff ≥ 10 reads in at least two of the biological replicates) in the metatranscriptome, and around 6% were detected in the metaproteome (exact numbers given in Fig. S10). Only a small portion of the entire metatranscriptome was significantly up/downregulated during the time series (Wald test, <0.05 $p$-value, >2-fold change) compared to t0, corresponding to 0.4/0.9% of the transcripts for the first time point, 4.7/5% for the second, and 5.4/5.4% for the third. The bins which were highly abundant in the metagenome (bins 10, 17, 20, and 32) contributed with the highest amount of upregulated genes at this time point. Bin 17 and 20 showed continued transcript upregulation after one week, while bin 10 and 32 had a reduced number of upregulated transcripts at the last time point. For each time point, the downregulated transcripts were mainly contributed by low abundance members of the I1 community (Fig. 4a).

In total 8126 protein groups were identified from the metaproteome, from which the highest amount of proteins (67%) were detected only after 7 days of incubation in the presence of new polymer films. The remaining time points showed a core of protein groups (≥3049, 38%) present at every time point. The highest number of upregulated protein groups (921 proteins) was consistently present after 7 days (Fig. S10).

We found that the most active bacteria in terms of protein biosynthesis were Alphaproteobacteria and Gammaproteobacteria throughout the entire experiment, especially of the genus *Marinobacter*. Between these two bacterial groups, the number of upregulated protein groups at every time point was greater for Gammaproteobacteria than for Alphaproteobacteria (Fig. 4a).

The addition of PF after a starvation period triggered the upregulation of several genes related to central metabolism, supporting that this substrate is used as carbon source. In addition, biofilm formation-related pathways (ko02025) were upregulated on the transcript level at the first and last time points, but not on protein level. Bacterial secretion systems (ko03070) showed increased transcription and translation (Fig. S11), which may be due to an increase in the production of secreted proteins such as Ples and Mles.

The *ple628*, *ple629*, and *ple453*, are the only *ple* genes that whose expression significantly increased in the metatranscriptome. The increase happened exclusively at the second time point. Consequently, these were also the only Ples detected in the metaproteome. Ple628 and Ple453 were highly abundant and upregulated throughout the time series (Fig. 5). The remaining Ples were absent from the proteome and possibly did not play a significant role in the depolymerization of PF. Due to competition in biofilms it is possible that redundant and cost-intensive metabolic functions are halted to favor selection for syntrophy.

Similar to the Ples, only one Mle (*mle046*), was upregulated in the metatranscriptome at the second and third time point. Interestingly, Mle046 was the most abundant protein group of the I1 metaproteome throughout the time series. It was also the only Mle protein that could be detected (Fig. 5).

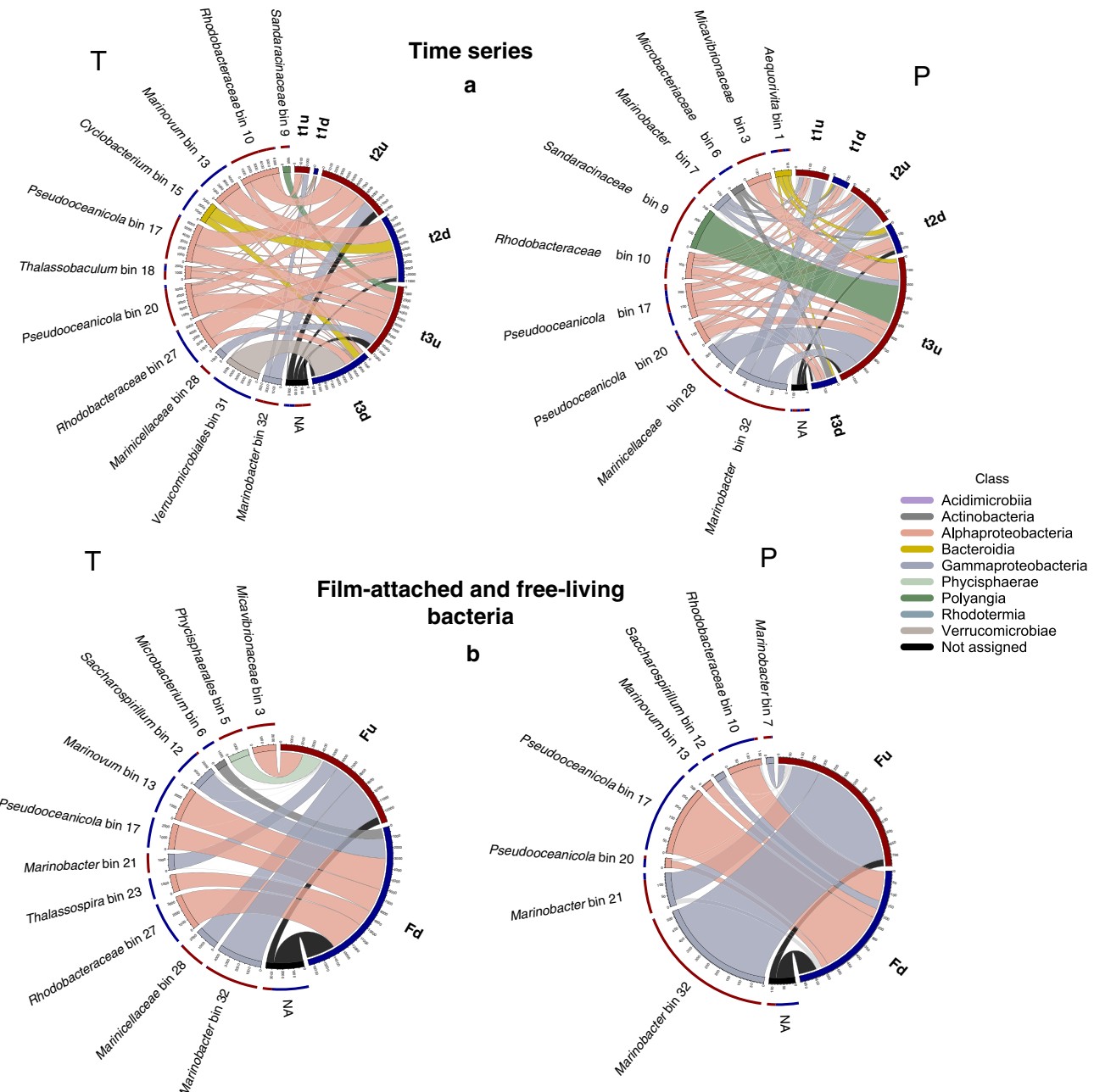

**Fig. 4 Bin contribution to gene expression and protein biosynthesis. a** The amount of significantly upregulated ("-u", red segment) and downregulated ("-d", blue segment) transcripts (T) and proteins (P) are shown for sampling time points (t1, t2, t3). **b** The amount of transcripts (T) and proteins (P) significantly upregulated in the film-attached community/downregulated in the free-living community ("Fu", red segment) or downregulated in the film-attached community/upregulated in the free-living community ("Fd", blue segment) are also shown. The average of at least two biological replicates were used to calculate transcript and protein fold changes. The amount of genes and proteins detected are represented by the size of each ribbon. Each color represents transcripts and proteins that belong to a certain bacterial class. The outer arc shows the amount of transcripts and proteins up/downregulated per bin (left side of each circle). Tick marks indicate numbers of transcripts and proteins up/downregulated per bin. NA represents genes that could not be allocated to any bin. Only bins that contributed with ≥1000 transcripts and ≥25 proteins are shown. Source data are provided as a Source Data file.

Both TPD clusters on contig 1092 (*ptpd*) and bin 12 (*stpd*) were transcribed during PF degradation. The *stpd* genes were significantly upregulated at the third time point, while the *pht* genes of bin 12 were upregulated at the first time point. Only sTphA2 and sTphA3 proteins were present, whereby the latter protein was downregulated at the last time point (Fig. 5). The pht proteins were not detected. The *ptpd* gene expression did not significantly change during the last two time points. The pTPD proteins, like Mle046, were highly abundant in the metaproteome at all time points without significant protein fold changes. Only

six PCD subunits were detected in the proteome. These belonged to bins 10, 17, and 20, as well as two unbinned subunits whose nearest relatives originate from Alphaproteobacteria (Fig. S12). Generally, the transcription of the subunits was upregulated throughout the time series while the protein biosynthesis was downregulated. No PCD proteins from bin 12 were detected.

**Gene expression and protein biosynthesis: film vs. free-living bacteria.** The taxonomic composition, gene expression, and

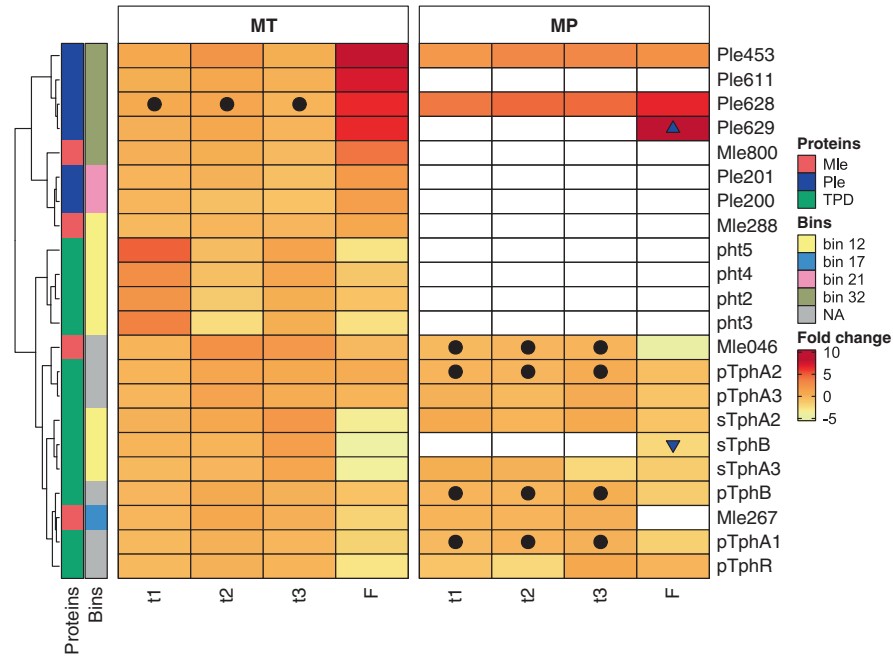

**Fig. 5 Heatmap of PF degradation genes and proteins.** Fold change profiles of genes potentially involved in PF biodegradation at the metatranscriptome (MT) and metaproteome (MP) level during different time points t1, t2, t3 and in the film-attached (F) community. The fold changes in (F) equal to the same amount of fold change in the free-living community with the opposite sign. The average of at least two biological replicates were used to calculate fold changes. Ples (blue): *ple453, ple628, ple611, ple629, ple201, ple200*; Mles (red): *mle046, mle267, mle800, mle288*; TPD cluster genes (green): *stphA2A3B* and *ptphA2A3BA1R*, and pht cluster genes: *pht3524*. Filled circles represent non-differential gene expression and protein biosynthesis along different time points. Blue-filled triangles indicate proteins unique in the film-attached fraction (up-pointing triangle) and unique in the free-living fraction (down-pointing triangle). The scale shows log2-transformed fold changes. Upregulated transcripts and proteins are shown in shades of red. Downregulated transcripts and proteins are shown in shades of yellow. Bins are represented by different colors. Samples were clustered according to the highest fold-change in the metatranscriptome. Source data are provided as a Source Data file.

protein biosynthesis profiles of film-attached and free-living community were significantly different. About 10% of all transcripts and 5.4% of the proteins were upregulated in the film-attached community, while 12% of the transcripts and 5% of the proteins were upregulated in the free-living community. In total, about 25%, 28%, 16% of the genes/transcripts/proteins respectively were unique to the free-living community, and 11%, 13%, 16% unique to the film-attached community (Fig. S10). Most transcripts of the film-attached community were contributed by bins 32, 28, 21, and 5, representing both high and low-abundance members (Fig. 4b). Yet, the majority of the proteins in the film-attached community were however produced by bin 32, and to a lesser extent by bin 21, the most abundant members. For the free-living community, transcript contribution from the high-abundance members bin 10, 17, and 21 were lower than their contribution to protein biosynthesis. The lower-abundant bins 12 and 13 were active in this fraction, contributing to both gene expression and protein biosynthesis. We have observed that pathways related to biofilm formation, bacterial secretion, quorum sensing and flagellar assembly were enriched in the biofilm metatranscriptome, but not in the metaproteome. Rather, the metaproteome showed an enrichment of fatty acid metabolism proteins, which may be related to the metabolism of dicarboxylic acids via β-oxidation[34–36] (Fig. S11).

The expression of all *ple* genes was restricted to the film-attached community. The *ple453* and *ple628* genes were upregulated in the film-attached community compared to the free-living fraction, and consequently more proteins were synthesized in the biofilm (Fig. 5). Similar to the time series experiment, neither Ple200 nor Ple201 could be detected in any of the fractions. Ple629 was detected in the film-attached community, in contrast to the time series experiment where this protein

could not be detected. The abundance of bin 32 in the biofilm, its metabolic potential, and gene expression profile suggests that this species could be the PF depolymerizing species in the consortium. The *mle046* gene was highly expressed in both film-attached and free-living communities without a significant difference in expression levels, however, more Mle046 was detected in the free-living community (4-fold). Other Mle gene expression levels were low and proteins were not detected. The *ptpd* transcripts and proteins were present in both film-attached and free-living communities. Their expression and biosynthesis levels were either similar in both fractions or upregulated in the free-living community. On the other hand, the *stpd* and *pht* transcripts were upregulated in the free-living community. Consistent with the time series experiment, only sTph and not pht proteins were detected. These were upregulated or only present in the free-living community (Fig. 5). Only two PCA dioxygenase subunits were detected in the proteomes during this experiment. These were not significantly up or downregulated in the film or free-living community (Fig. S12). Although the transcripts of PCA dioxygenase subunits from bins 10 and 12 were upregulated in the film community; these were not detected in the proteome.

## Discussion

In this study, we used a multi-omics approach to elucidate the biodegradation potential of a commercial aliphatic-aromatic polyester blend by a marine microbial consortium. Our data shows that both the polymers and its monomers can be mineralized by the consortium, and points towards a synergistic biodegradation process within the I1 community. Labor division within microbial consortia to completely degrade complex anthropogenic chemicals has been demonstrated before[37–39]. In

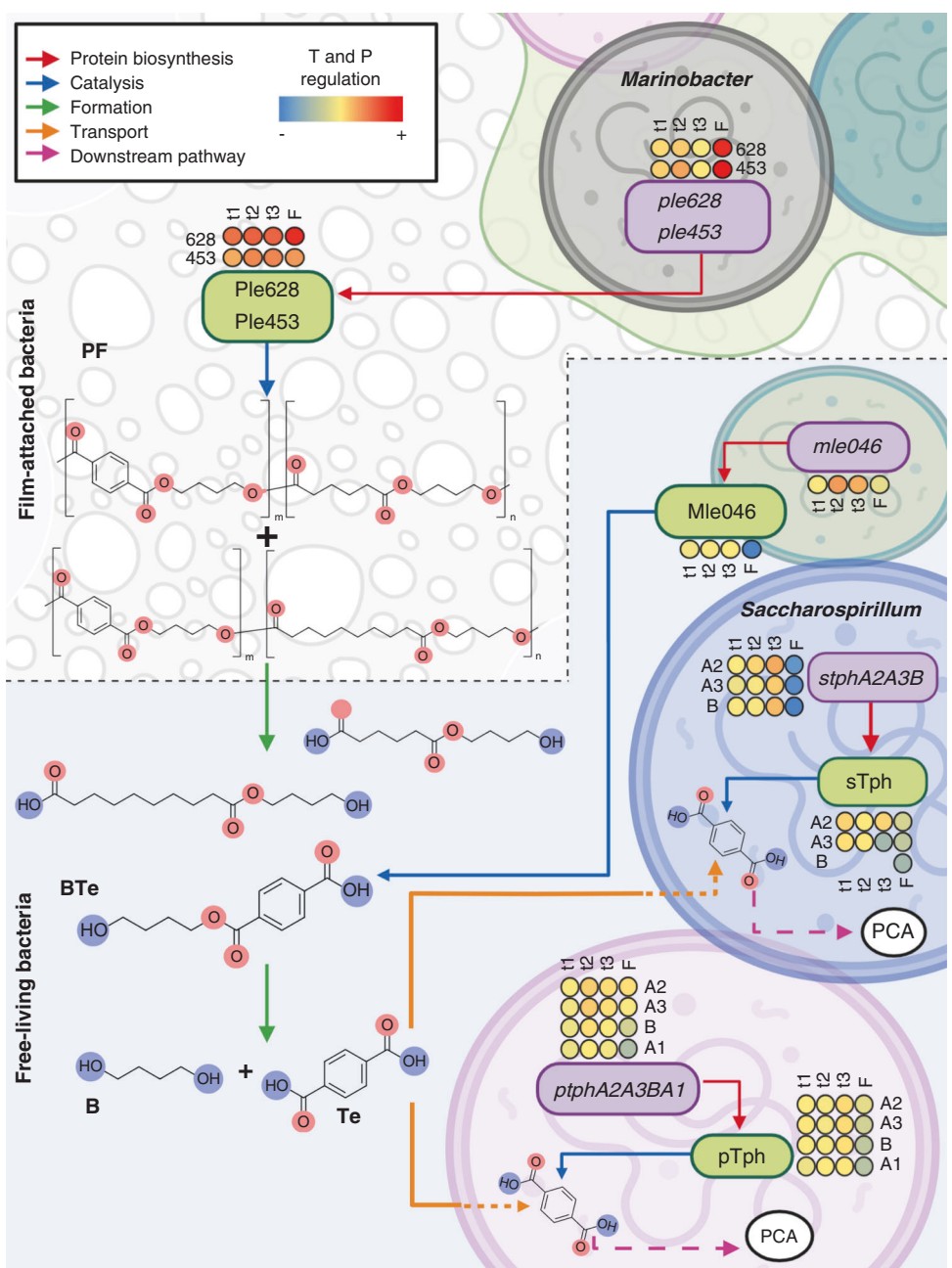

**Fig. 6 Integration of multi-omics data.** Proposed mechanisms for the synergistic biodegradation of a PBAT-based blend film (PF) during different time points (t1–t3) and within a biofilm (F). Periplasmatic $\alpha/\beta$ hydrolases (Ple628, Ple453), synthesized by *Marinobacter*, hydrolyze PF to produce oligomers and monoesters of 1,4-butanediol (B) with either terephthalic acid (Te) or sebacic acid (Se)/adipic acid (Ad). An Alphaproteobacterial Mle (Mle046) hydrolyzes a mixture of terephthalate-butanediol monoester (BTe) to produce Te and B. Te is transported into the cell and catabolized by terephthalate degradation clusters of an unknown bacterium (pTPD), and to a lesser extent by *Saccharospirillum* (sTPD). Downstream degrading genes leads to the formation of protocatechuate (PCA). Red arrows indicate gene translation into a functional protein; blue arrows indicate the catalysis of a substrate by a protein; green arrows, show the formation of intermediates by the catalysis of a substrate; orange arrows represents the transport of intermediates from the surroundings to bacteria; pink arrows represent donwstream degradation pathways. The color scale represent upregulated (+) and downregulated (−) transcripts (T) and proteins (P) (created with Biorender.com).

the I1 culture, different members of the community perform depolymerization, breakdown of intermediates, and the aromatic monomers as proposed in Fig. 6.

The bacterial community on PF is comparable to what has been found on biofilms colonizing other plastics particles and is mostly comprised of *Gammaproteobacteria*[40,41]. Some members of this class are early colonizers of marine substrates[40,42] and have been found colonizing plastics[43–47]. The data suggests that, among the Gammaproteobacteria, the genus *Marinobacter* plays

a crucial role in PF degradation. The hydrocarbonoclastic capability of some *Marinobacter* species[48–50], as well as their abundance on the film and their corresponding activity point towards this potential. The genomes of both *Marinobacter* bins (bin 21 and 32) encoded for Ples. Bin 32 possibly performs the initial depolymerization of PF (Fig. 6), allowing access to the rather hydrophobic substrate[40]. Bin 32, which lacks Te degradation genes, was abundant when the culture received the dicarboxylic acid monomers as sole carbon source. Moreover, pathways

related to the metabolism of dicarboxylic acids were enriched in the biofilm metaproteome where this bacterium is abundant. Thus, it is likely that after depolymerization, this microorganism grows on the available dicarboxylic acid monomers. A cultivation-dependent approach may not accurately represent how the degradation process would occur in the environment, however our results suggest that the marine aliphatic-aromatic polyester degradation genes are highly similar to those identified in terrestrial microorganisms.

Within the free-living fraction, Alphaproteobacteria comprised the highest abundance, mainly by members of the Rhodobacteraceae family. Rhodobacteraceae thrive in marine environments[51,52] and several of them can alternate between free-living and attached lifestyles[53–55]. This lifestyle is especially apparent for bins 10 and 20, which are equally abundant in biofilm and supernatant. Alphaproteobacteria in the I1 community are therefore likely to feed on the soluble oligomers and monomers after depolymerization. The highly-expressed mle and tpd genes belong to Alphaproteobacterial hosts, as well as the majority of the pcd genes. We therefore assume that Alphaproteobacteria play a key role in the mineralization of aromatic oligmers and monomers.

We have observed that bins 12 and 7, which outgrow the other community members when Te and B are given as sole carbon source, have a low abundance (<2%) when the community is growing on PF. After PF cleavage by the Ples, B and Te could be available as the monoester BTe at first rather than as free monomers, similar to the release of MHET by IsPETase[18]. The intracellular presence of BTe during the PF degradation further indicates that this monoester is taken up and used as a C source rather than being cleaved into the monomers extracellularly first (see proposed alternative mechanism in Fig. S13). Bins 7 and 12 are thus outcompeted by mle046-encoding organisms when the culture is growing on PF. Since both mle046 and ptpd genes are located on plasmids, it is hard to state exactly which community members use BTe as growth substrate. Bins 10 and 17 are very abundant when the community is growing on the PF, but become nearly undetectable when monomers are supplied as C source. Based on this observation, we postulate that either one or both of these bacteria are the hosts of the mle046 and ptpd-carrying plasmids. These bacteria possibly take up and use BTe as a C source, but are outgrown by bin 12 and bin 7 when monomers Te and B are freely available. When the community grows on PF as the C source, the bin 12 and 7 possibly exist as secondary degraders.

Taxa with lower abundance also showed an increase in activity throughout the degradation process. Downstream degradation products and leaking metabolites could be metabolized by lower abundant taxa such as bin 9 (Sandaracinaceae), which contributed most of the upregulated proteins at the last sampling point. Members of this family consume low molecular weight compounds in marine ecosystems (i.e., acetate, ethanol)[56].

Based on the gene expression and protein biosynthesis patterns of the depolymerases and downstream degradation genes, we conclude that PF depolymerization is performed by the film-attached community. Aromatic oligomers and monomers are degraded by both the film and free-living community, thus sharing the task of complete PF mineralization.

## Methods

**Inoculum source.** Marine seawater and eulittoral sediment samples were collected from three different sources: Helgoland, Germany (54° 11' 17.0" N 7° 52' 50.1" E), near Athens, Greece (37° 53' 33" N, 23° 24' 30" E) and Elba, Italy (42° 44' 00.8" N 10° 09' 14.4" E) (November 2016). Samples were stored in the dark at 4 °C until further use.

**Reagents.** PF (ecovio® FT 2341) was supplied by BASF (Ludwigshafen, Germany). The monomers were purchased from Sigma Aldrich (Missouri, USA) with the highest purity grade available.

**Culture conditions.** All cultivations were performed in a mineral media as described before for *Pirellula* (600a. M13a DSMZ medium)[57] with some modifications: Artificial sea water (ASW 1X), Hutner's salts and metals solutions were prepared according to DSMZ 590 medium. The carbon and nitrogen sources described for these media (600a. M13a and 590 DSMZ media) were omitted and instead NH4Cl (1 mM) and K2HPO4 (0.1 mM) were used. The pH of cultures supplied with either Se or Ad were maintained with HEPES buffer instead of bicarbonate.

The enrichment cultures were set up with 50 mL ASW medium supplemented with 10 mg of PF and 0.01% tryptone. To prepare the inoculum, 1 g of each sediment and 10 mL of each seawater sample were pooled, vortexed for 2 min and allowed to settle. One mililiter of the supernatant was used to inoculate each culture. After film disintegration was observed (ca. two months), 1 mL of the liquid culture was transferred into fresh ASW medium with 10 mg of PF, but without the addition of tryptone. After four transfers, an enrichment culture capable of disintegrating the plastic film in a time frame of six days was obtained and named I1. The culture was routinely maintained in ASW medium with PF as carbon source. Cultivations were performed at pH 7.0 and incubated in a MaxQ™ 4000 orbital shaker (Fisher Scientific, Schwerte, Germany) at 22 °C, 120 rpm.

To measure copolymer and monomer mineralization, the I1 community was inoculated in 25 mL of mineral media containing 50 mg of carbon per liter of either PF, Te, B, Ad, or Se. Each flask had a volume of 250 mL and contained an external $CO_2$ trap. Copolymer or monomer mineralization were measured from three biological replicates. In addition, three negative controls consisting of cultures without a carbon source were also included in the experiment. All cultivations were performed at pH 7.0 and incubated in a MaxQ™ 4000 orbital shaker (Fisher Scientific, Schwerte, Germany) at 22 °C, 120 rpm. Cultures were harvested for metagenomics in stationary phase after the carbon source was depleted. Cultures were centrifuged in a Heraeus™ Multifuge™ X3 (Fischer Scientific, Schwerte, Germany) at 4 °C, 8000 rpm for 10 min.

Differences between film-attached communities and free-living bacteria were analyzed in 200 ml of mineral media supplied with 5 × 5 cm PF. The experiment consisted of three biological replicates that were incubated in a MaxQ™ 4000 orbital shaker (Fisher Scientific, Schwerte, Germany) at 22 °C, 120 rpm. Before disintegration of the PF, the two fractions of the bacterial communities were separated by filtration. The filtrate was consequently centrifuged in a Heraeus™ Multifuge™ X3 (Fischer Scientific,Schwerte, Germany) at 4 °C, 8000 rpm for 10 min. The films and recovered pellets were flash-frozen in liquid nitrogen and stored at −80 °C until DNA, RNA and protein extraction.

Gene expression and protein biosynthesis were analyzed in response to the presence of PF in a time series experiment. For this, we used 67.2 ± 0.1 mg of PF in 200 mL of mineral media. The incubation was carried out in three 2 L baffled flasks in a MaxQ™ 4000 orbital shaker (Fisher Scientific, X, Germany) at 22 °C, 70 rpm (instead of 120 rpm). Cultures were harvested during starvation (control: t0), immediately after 1 h of PF addition (t1), after 24 h (t2), and after 7 days (t3). The last sample was taken right before the disintegration of the film. At each time point, 30 mL of a free-living and attached biofilm mixture was extracted and centrifuged in a Heraeus™ Multifuge™ X3 (Fischer Scientific, Schwerte, Germany) at 4 °C, 8000 rpm for 10 min. The bacterial pellets were flash frozen in liquid nitrogen and stored at −80 °C until DNA, RNA, and protein extraction.

For all experiments, cultures previously grown with PF as the pre-inoculum were used.

**Bacterial growth and $CO_2$ production.** The extent of mineralization was calculated based on $CO_2$ formation. The $CO_2$ formed in the mineralization tests were measured as inorganic carbon using a TOC-L analyzer and data collection was retrieved by using the TOC-Control L v. 1.06 software (Shimadzu, Kyoto, Japan). Before the analysis, the released $CO_2$ was entrapped in 4 mL of 0.1 mM NaOH to form NaHCO3 and diluted 10× with Milli-Q water. The carbon content was determined using a calibration curve of NaHCO3.

**Detection of PF residues and byproducts (FTIR; GC-MS).** Residues originated from PF mineralization tests were analyzed by FTIR using a PerkinElmer Spectrum 100 FT-IR Spectrometer (PerkinElmer, Massachusetts, USA). The scans were performed from 4000 to 650 $cm^{-1}$ with a resolution of 2,00 $cm^{-1}$. The resulting spectra were visualized and analyzed with Spectrum Quant software v. 10.4 (PerkinElmer, Massachusetts, USA). Figures were further edited with the software Spectragraph v. 1.2.14 (https://www.effemm2.de/spectragryph/).

The monoester BTe was used as a GC-MS standard in addition to the monomers. This compound was synthesized following the procedure described by Perz et al.[58]. The identity was confirmed with ¹H NMR obtained on a Bruker Avance III platform. The purity was estimated to be 93%. 1H NMR (500 MHz, DMSO-d6) δ 13.34 (s, 6H), 8.06 (s, 28H), 4.45 (s, 6H), 4.38 (s, 2H), 4.31 (t, J = 6.6 Hz, 13H), 3.89 (q, J = 1.0 Hz, 1H), 3.45 (d, J = 12.9 Hz, 13H), 3.31 (s, 3H), 1.93–1.87 (m, 1H), 1.82–1.67 (m, 13H), 1.61–1.50 (m, 13H).

Metabolites were extracted from the bacterial pellets as described before[59] with minor modifications. Briefly, cell pellets were resuspended in 500 µL methanol (containing 5 mg/mL 13C5-ribitol as internal standard) and cells were lysed in an ultrasonic bath at 70 °C for 15 min. The same volume of water was added, the sample was mixed and centrifuged for 5 min at 12,000×$g$ to remove residual plastic film/particulate material. The supernatant was vigourously mixed with 500 µL of chloroform, centrifuged for 5 min at 17,000 × $g$ and 800 µL of the polar phase were dried in vacuum. GC-MS analysis was performed on an Agilent GC-MSD system (7890B coupled to a 5977 GC) equipped with a high-efficiency source (HES) and a PAL RTC system using a two-step derivatization with a methoxyamine hydrochloride solution (20 mg mL$^{-1}$ in pyridine) and N-methyl-N-(trimethylsilyl)-trifluoroacetamide. The autosampler was operated by the Maestro 1.5.4.2/3.5 (Gerstel) software. Targeted GC-MS analysis was performed on an Agilent GC-MSD system (7890B coupled to a 5977 GC) (Agilent Technologies, California, USA) equipped with a high-efficiency source (HES) and a PAL RTC system as previously described[59] using authentic standards of adipic acid, sebacic acid, terephthalic acid and BTe. Data was analyzed with the MassHunter GC/MS Acquisition B.07.06.2704 (Agilent Technologies, California, USA).

**Field emission scanning electron microscopy (FESEM)**. PF-attached bacteria were fixed with 5% formaldehyde and 2% glutaraldehyde in HEPES buffer (0.1 M HEPES, 0.01 M CaCl$_2$, 0.01 M MgCl$_2$, 0.09 M sucrose, pH 6.9) on ice, then washed twice with TE buffer (20 mM TRIS, 2 mM EDTA, pH 6.9). The treated samples were then dehydrated with graded series of ethanol (10, 30, 50, 70, 90, 100%) on ice for 10 min for each step and allowed to reach room temperature before another change in 100% ethanol. Consecutively, samples were then subjected to critical-point drying with liquid CO$_2$ (CPD 030, Bal-Tec AG, Balzers, Liechtenstein). Dried samples were covered with an approximately 8 nm thick gold-palladium film by sputter coating (SCD 500 Bal-Tec, Balzers, Liechtenstein) before examination in a field emission scanning electron microscope Zeiss Merlin (Carl Zeiss AG, Oberkochen, Germany) using the Everhart-Thornley SE-detector and the Inlens SE-detector in a 75:25 ratio with an acceleration voltage of 5 kV. Images were stored digitally with SEMSmart software v. 5.05.

**DNA, RNA, and protein extraction, purification and quantification**. DNA, RNA and proteins were extracted from most of the experiments as shown in the experimental setup (Fig. 1). Only DNA was extracted from the monomer mineralization tests. Total DNA, RNA and proteins were extracted from all tests simultaneously by using the Quick-DNA/RNA Miniprep Plus Kit (Zymo Research, California, USA) according to the manufacturer's instructions. To asses the integrity of isolated DNA and RNA, 5 µL of each sample was loaded in an agarose gel and visualized by using Intas Gel v. 0.2.14 software. To remove possible RNA contamination, the DNA samples were additionally treated with RNase A (AppliChem, Darmstadt, Germany) and purified by using DNA Clean and Concentrator kit (Zymo Research, California, USA) following manufacturer's instructions. Quality of DNA and RNA was additionally analyzed via Nanodrop by using the Nanodrop 2000 v. 1.5 software (Nanodrop 2000, Thermo Fischer Scientific, Waltham, USA). The fractions of proteins isolated with Quick-DNA/RNA Miniprep Plus Kit (Zymo Research, California, USA) were further precipitated with acetone according to the manufacturer's instructions.

Purified total DNA, RNA, and proteins were quantified to determine concentration by using Qubit RNA BR, dsDNA BR, and Protein Assay kits and measured on a Qubit 3.0 Fluorometer (Invitrogen, California, USA).

**DNA and RNA sequencing**. The DNA sequencing library was generated from 200 ng DNA using NEBNext Ultra II FS DNA Library Prep Kit for Illumina (New England BioLabs, Massachusetts, USA) according to manufacturer's protocols including PCR amplification with four cycles. The libraries were sequenced on Illumina NovaSeq 6000 using the NovaSeq 6000 S1 PE Reagent Kit (300 cycles) (Illumina, California, USA) with an average of 1E7 reads per DNA sample. Data was analyzed and converted to FASTQ files using the NovaSeq Control Software v1.6, RTA Version 3.4.4 and bcl2fastq v2.20.0.422.

Quality and integrity of total RNA was controlled on Agilent Technologies 2100 Bioanalyzer (Agilent Technologies, Waldbronn, Germany). The RNA sequencing library was generated from 250 ng total RNA using Ribo-off rRNA Depletion Kit (Bacteria) (Vazyme BioTech, Nanjing, China) for rRNA depletion followed by NEBNext® Ultra™ II Directional RNA Library Prep Kit (New England BioLabs, Massachusetts, USA) according to the manufacturer's protocols. The libraries were sequenced on Illumina NovaSeq 6000 using NovaSeq 6000 S1 PE Reagent Kit (100 cycles) (Illumina, California, USA) with an average of 2E7 reads per RNA sample.

**Metaproteomics by nano LC-MS/MS**. The protein precipitates were dissolved in SDS-PAGE sample loading buffer, loaded on an SDS-gel and run for 10 min. The gel pieces were cut, washed and incubated with 25 mM 1,4-dithiothreitol (in 20 mM ammonium bicarbonate) for 1 h and 100 mM iodoacetamide (in 20 mM ammonium bicarbonate) for 30 min. The pieces were further destained, dehydrated and proteolytically cleaved overnight at 37 °C with trypsin (Promega, Walldorf, Germany). The digested peptides were extracted and desalted using ZipTip-C18 tips (Merck Millipore, Darmstadt, Germany). Afterwards, the peptide lysates were re-suspended in 0.1% formic acid and injected to a nanoliquid chromatography mass spectrometry (nanoLC-MS/MS).

Mass spectrometric analysis of peptides was performed on a Q Exactive HF mass spectrometer (Fisher Scientific, Massachusetts, USA) coupled with a TriVersa NanoMate (Advion, Ltd., Harlow, UK) source in LC chip coupling mode. LC gradient, ionization mode and mass spectrometry mode have been used as described before[60].

Data resulting from LC-MS/MS experiments were analyzed using the Proteome Discoverer v. 1.4 (Fischer Scientific, Massachusetts, USA) using SEQUEST HT. As a database, the protein-coding sequences of the metagenome were used. Search settings were set to trypsin (Full), max. missed cleavage: 2, precursor mass tolerance: 10 ppm, fragment mass tolerance: 0.02 Da. The false discovery rates (FDR) were determined with the node Percolator[61] embedded in Proteome Discoverer and the FDR threshold was set at a peptide level of <1%. The same threshold was set for the protein FDR.

**Data analysis**. The metagenomes were processed with MetaWRAP v. 1.2[62]. In brief, the reads were trimmed with Trim Galore! v. 0.6.4[63] and overrepresented sequences comprising technical artifacts were filtered out. All reads were co-assembled with MetaSPAdes v. 3.13.0[64] which yielded in 75,703 contigs (174.8 Mbp with a N50 of 118,039). Gene calling and annotation was performed on the metagenome with Prokka v. 1.14[65]. Binning of the assembled sequences was carried out with CONCOCT v. 1.0.0[66], MaxBin v. 2.2.5[67] and Metabat v. 2.12.1[68], which were subsequently refined with MetaWRAP. Completeness and contamination of the refined bins was assessed with checkm v. 1.0.12[69], the taxonomy of the bins was assigned with GTDBtk v. 0.3.2[70]. The trimmed reads were mapped on the assembled metagenome with the Subread aligner[71] from the RSubread package v. 1.34.7[72]. Abundance estimation was performed with samtools v. 1.7[73]. The correlation of the relative abundance of the bins was visualized with the ggcorrplot package v. 0.1.3.999[74] using Pearson correlation with a p-value significance cutoff of 0.05 and hierarchical clustering.

Metatranscriptomics reads were trimmed with Trim Galore! v. 0.6.4[63] and rRNA reads were removed with SortMeRNA v. 2.1b[75]. Subsequently, the remaining reads were mapped on the assembled metagenome the Subread aligner[71] from the RSubread package v. 1.34.7[72] (Table S5). Differential gene expression analysis was conducted with DeSEQ2 v. 1.24.0[76]. In brief, genes were normalized with the median of ratios method[77]. The differential gene expression analysis was performed by fitting the negative binomial model for each gene and subsequent hypothesis testing with the Wald test. The shrunken log2 foldchanges (LFC) were calculated using the adaptive t prior shrinkage estimator with apeglm v. 1.6.0[78]. The p-values were adjusted for multiple testing using the Benjamini and Hochberg method[79]. A gene is considered to be significantly differentially expressed with ≤two-fold change and an adjusted $p$ value < 0.05 with a false discovery rate (FDR) < 0.05.

For metaproteomic quantification the redundant proteins were grouped in protein groups by applying the strict parsimony principle. Only the protein groups that explain at least one unique identified peptide were reported. Only the peptides that were not shared between different proteins or protein groups were used for the protein quantification through the Top3 approach implemented in the Proteome Discoverer v. 1.4 (Fischer Scientific, Massachusetts, USA). The protein data were further log2 transformed and normalization of the peptides were manually performed by dividing the transformed values by the median of the sample and multiplying by the mean of entire data set[80,81] with Microsoft Excel 2010 v. 14.0.7252.5000.

For both comparative metatranscriptomic and proteomic analyses, fold changes were calculated relative to t0 and free-living fraction for the time series and film-attached communities analysis respectively.

The replicate consistency was determined by analyzing the distribution of the normalized counts (for MT) and normalized area (for MP), respectively (Fig. S14a). Unsupervised clustering of the metatranscriptomes and metatranscriproteomes was done and visualized with ComplexHeatmap v. 2.1.0[82] (Fig. S14b). The input data was the normalized count numbers of each biological replicate. Circos plots were created with the online tool Circos Table Viewer[83]. KEGG over-representation tests were carried out and visualized using ClusterProfiler v. 3.14.3[84]. All R packages were used with R v. 3.6.0[85]. All tools were run with default parameters.

Protein alignments were performed with T-Coffee online platform[86] (http://tcoffee.crg.cat/apps/tcoffee/do:regular, last visited 30.09.2020) and similarities were shaded with BOXSHADE v. 3.2 online source (https://embnet.vital-it.ch/software/BOX_form.html). Phylogenetic trees were constructed for Ples, Mles and TPADO based on their aminoacidic sequences and compared to other homologous proteins. For this, the sequences obtained in this study together with homologous sequences were aligned with ClustalW v. 2.1[87] by using the Geneious® Prime v.2019.1.1 software (https://www.geneious.com). BLOSUM was selected as the scoring matrix and gaps were set at 10 (gap open cost) and 0.1 (gap extend cost) per element. The trees were generated with Geneious® Prime v. 2019.1.1 (https://www.geneious.com) by using the Jukes-Cantor distance model and built with the Neighbor–Joining method[88]. No outgroups were included in any of the phylogenetic trees. The number of bootstrap replicates were 1000. Newick phylogenetic trees were further midpoint rooted and edited with iTOL v. 5.5[89].

**Reporting summary**. Further information on research design is available in the Nature Research Reporting Summary linked to this article.

## Data availability

The metagenomic and metatranscriptomic raw data have been deposited at EBI Metagenomics/MGnify: PRJEB37199, the metaproteomics data at the ProteomeXchange Consortium via PRIDE with the identifier PXD018391. KEGG (https://www.genome.jp/kegg/pathway.html) and Uniprot (https://www.uniprot.org/) databases were used for protein annotation. All other data are available from the corresponding author upon request. Source data are provided with this paper.

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

## Acknowledgements

The authors thank Anja Heuer (Microbial Biotechnology group, DSMZ) and Gesa Martens (Bacterial Metabolomics group, DSMZ) for technical assistance, Kathleen Eismann (Molecular System Biology department, UFZ), and Silke Reinecke (Natural Product Chemistry group, HZI), for providing us with technical expertise and support. Manfred Rohde (Central Facility for Microscopy, HZI) is acknowledged for the generation the SEM images, and Mark Brönstrup (Chemical Biology department, HZI) is acknowledged for providing access to the FTIR spectrometer. In addition, the authors acknowledge the use of de.NBI cloud and the support by the High Performance and Cloud Computing Group at the Zentrum für Datenverarbeitung of the University of Tübingen and the Federal Ministry of Education and Research (BMBF) through grant no 031 A535A. The authors thank BASF SE for financial support.

## Author contributions

I.E.M.C. designed and performed experiments, analyzed the data, and wrote the paper; J.W. carried out the bioinformatic analysis of the metagenomes and metatranscriptomes; N.J. performed the metaproteomic analysis by nano LC-MS; M.N. and S.W. performed the metabolite analysis by GC-MS; B.Ö designed the experiments, analyzed the data, wrote the paper, and supervised the project. All authors have contributed to the editing of the paper and agree on the final version.

## Funding

## Competing interests

The authors declare no competing interests.
