## [Peer Review File · Nature Communications]

REVIEWERS' COMMENTS

Reviewer #2 (Remarks to the Author):

The article "Synergistic biodegradation of a poly(butylene adipate-co-terephthalate) derivative by a marine microbial consortium" is reporting wet-lab experiments and derived data analyses of a microbial enrichment able to degrade PBAT-based blend film.

This article has already been reviewed for another submission and this assessment is both based on the referee reading of the current version of the manuscript and the answers of the authors to previous comments (albeit the referee couldn't track the changes that have been between the 2 versions of the manuscript).

The manuscript reads well and is concise, figures illustrate well the scientific message (except maybe for figure 3a, see below). Comments from all previous reviewers were diligently taken into account.

COMMENTS:

"CO₂ evolution" is used a few time in the text and supplemental materiel. CO₂ is not evolving, but [CO₂] (its concentration) is changing. Please, rephrase these few sentences to make clear that you are mentioning CO₂ emission/production or [CO₂] dynamics.

The section 2.2 would deserve a very small introduction, maybe just (i) mentioning that the text is now about the experiments introduced in Fig 1b and 1c (why not re-citing this panel at appropriate places?); and (ii) a sentence introducing that DNA (or biomolecules) was (were) collected, sequenced and assembled before directly describing bins.

Twice (lines 148-149 and 166), the authors put a sentence like "[...] bins comprise xx% of the binned populations". Do they mean that xx% of the aligned reads map to these bin? How much reads do not map back to the bins? This information is important, especially because unbinned contigs are also analyzed when searching for homologs of known enzymes.

Albeit being very informative, Figure 3a is difficult to interpret on a A4 hardcopy or without an extensive zooming on screen. Most of the bubbles seems to be of the same size, but authors comment on some of them to 'double their population" (I cannot see that bins 6, 12, 14, 18 and 26 double their population size in S compared to F..., and bin 4 and 5 for the reverse comparison!). Would another scaling or display help the authors to support the related text, that is very precious to this article?

Also, in the related sentence line 155 "...doubled their population in the free-living fraction...", please mention that you are not describing absolute population size as it currently reads, but relative abundance/relative population size.

Figure 4 is missing the information of which plot is describing transcriptomic and which one is describing proteomics.

Figure S9 is supposed to use a gradient of blue for show significance, but most of the bubble are light grey. Do that mean that they are bellow a significance threshold? Then please add this information.

Reviewer #3 (Remarks to the Author):

The authors have considerably improved the manuscript according to the reviewers suggestions. Yet it is unfortunate, that all biodegradation data are presented for blended films only while no information is supplied about the individual components. It is correct that different physical properties of blends will affect biodegradation, - yet important information could be deducted related to the susceptibility of the different chemical bonds present in the different constituent polymers.

Reviewer #2 (Remarks to the Author):

The article "Synergistic biodegradation of a poly(butylene adipate-co-terephthalate) derivative by a marine microbial consortium" is reporting wet-lab experiments and derived data analyses of a microbial enrichment able to degrade PBAT-based blend film.

This article has already been reviewed for another submission and this assessment is both based on the referee reading of the current version of the manuscript and the answers of the authors to previous comments (albeit the referee couldn't track the changes that have been between the 2 versions of the manuscript).

The manuscript reads well and is concise, figures illustrate well the scientific message (except maybe for figure 3a, see below). Comments from all previous reviewers were diligently taken into account.

Reply to the referee: We would like to thank for the reassessment of the manuscript and the constructive comments. Since the paper was transferred to Nature Communications, we treated it as a first submission, hence the MS did not contain tracked changes.

COMMENTS:

"CO₂ evolution" is used a few time in the text and supplemental materiel. CO₂ is not evolving, but [CO₂] (its concentration) is changing. Please, rephrase these few sentences to make clear that you are mentioning CO₂ emission/production or [CO₂] dynamics.

Reply to the referee: We have changed all instances of "evolution" to "production" for clarity.

The section 2.2 would deserve a very small introduction, maybe just (i) mentioning that the text is now about the experiments introduced in Fig 1b and 1c (why not re-citing this panel at appropriate places?); and (ii) a sentence introducing that DNA (or biomolecules) was (were) collected, sequenced and assembled before directly describing bins.

Reply to the referee: We have added a small introduction as suggested (lines 148-151). We have also added references to the corresponding panels of Figure 1 to enable the easier connection of the results to the experiments.

Twice (lines 148-149 and 166), the authors put a sentence like "[...] bins comprise xx% of the binned populations". Do they mean that xx% of the aligned reads map to these bin? How much reads do not map back to the bins? This information is important, especially because unbinned contigs are also analyzed when searching for homologs of known enzymes.

Reply to the referee: Yes, it means that the respective percentage of the reads map to this bin. We have added a supplementary table (Table S3) with the percentages of reads which do not map to any bin per replicate. The proportions of unbinned reads are generally low, ranging from 5-23%.

Albeit being very informative, Figure 3a is difficult to interpret on a A4 hardcopy or without an extensive zooming on screen. Most of the bubbles seems to be of the same size, but authors comment on some of them to 'double their population' (I cannot see that bins 6, 12, 14, 18 and 26

double their population size in S compared to F..., and bin 4 and 5 for the reverse comparison!). Would another scaling or display help the authors to support the related text, that is very precious to this article?

Also, in the related sentence line 155 "...doubled their population in the free-living fraction....", please mention that you are not describing absolute population size as it currently reads, but relative abundance/relative population size.

Reply to the referee: In order to increase the clarity of this figure, we decided to divide it into one main (3a) and two supplementary (S4 and S5) figures. The abundance profiles for the monomer culture experiments were moved to Figure S5, as these experiment had stark contrasts between the highest and lowest abundant taxa, which made scaling within one figure difficult. We have also added Figure S4, which illustrates a zoom-in of the bins with less than 1% abundance in Figure 3a. We hope that these changes will make the viewing and interpretation of this figure easier. We also reformulated the mentioned statement (lines 164-168) for clarity.

Figure 4 is missing the information of which plot is describing transcriptomic and which one is describing proteomics.

Reply to the referee: We added the missing headers to the figure.

Figure S9 is supposed to use a gradient of blue for show significance, but most of the bubble are light grey. Do that mean that they are bellow a significance threshold? Then please add this information.

Reply to the referee: Most circles in the figure appear gray, as the significance for most transcripts and proteins were below the significance threshold (0.05). This information was added to the legend.

Reviewer #3 (Remarks to the Author):

The authors have considerably improved the manuscript according to the reviewers suggestions. Yet it is unfortunate, that all biodegradation data are presented for blended films only while no information is supplied about the individual components. It is correct that different physical properties of blends will affect biodegradation, - yet important information could be deducted related to the susceptibility of the different chemical bonds present in the different constituent polymers.

Reply to the referee: We agree that the chemical susceptibility of the bonds in individual components can be important in understanding the degradation behavior. We think however, that this aspect is better addressed in enzyme assays rather than mineralization assays. We are planning to investigate this aspect in further publications.